# Periodontitis and Its Inflammatory Changes Linked to Various Systemic Diseases: A Review of Its Underlying Mechanisms

**DOI:** 10.3390/biomedicines10102659

**Published:** 2022-10-21

**Authors:** Ruchi Bhuyan, Sanat Kumar Bhuyan, Jatindra Nath Mohanty, Srijit Das, Norsham Juliana, Izuddin Fahmy Abu

**Affiliations:** 1Department of Oral Pathology & Microbiology, IMS and SUM Hospital, Siksha ‘O’ Anusandhan University (Deemed to be), Bhubaneswar 751003, India; 2Department of Medical Research, IMS and SUM Hospital, Siksha ‘O’ Anusandhan University (Deemed to be), Bhubaneswar 751003, India; 3Institute of Dental Sciences, Siksha ‘O’ Anusandhan University (Deemed to be), Bhubaneswar 751003, India; 4School of Applied Sciences, Centurion University of Technology and Management, Jatni, Bhubaneswar 752050, India; 5Department of Human and Clinical Anatomy, College of Medicine & Health Sciences, Sultan Qaboos University, Muscat 123, Oman; 6Faculty of Medicine and Health Sciences, Universiti Sains Islam Malaysia, Nilai 71800, Malaysia; 7Institute of Medical Science Technology, Universiti Kuala Lumpur, Kuala Lumpur 50250, Malaysia

**Keywords:** periodontitis, oral, infection, mechanisms, pathology, management

## Abstract

Periodontitis is a chronic inflammatory disease of the gums. The incidence of periodontitis is increasing all over the world. In patients with periodontitis, there is gradual destruction of the periodontal ligament and the alveolar bone, and later, in advanced stages, there is tooth loss. Different microorganisms, the host’s immune response, and various environmental factors interact in the progression of this chronic inflammatory disease. In the present review, we discuss the epidemiology, clinical features, diagnosis, and complications of periodontitis. We also discuss the association of chronic inflammation found in periodontitis with various other systemic diseases, which include cardiovascular, respiratory, diabetes, Alzheimer’s, cancer, adverse pregnancy, and multiple myeloma, and also highlight microbial carcinogenesis and the microRNAs involved. The latest updates on the molecular mechanism, possible biomarkers, and treatment procedures may be beneficial for diagnostic and therapeutic purposes.

## 1. Introduction

Periodontitis is a clinical condition where there is chronic inflammation of the periodontium, resulting in the loss of the periodontal ligament and damage to the surrounding alveolar bone. At first, the teeth loosen, and at an advanced stage, there may be tooth loss. Periodontitis occurs due to an imbalance in the oral microbiota’s natural balance and host resistance (dysbiosis), and has been linked to various systemic conditions. In adults, dysbiosis may be the cause of periodontitis, which can lead to inflammatory changes that affect the bone and connective tissue [1].

Periodontitis is one of the most common inflammatory conditions that involves the oral cavity and it has also been linked to cancer [2]. Bacterial plaque is the cause of periodontitis in susceptible individuals. Improper oral hygiene and tobacco consumption, combined with genetically- and disease-associated disturbances of host defenses, are the main factors associated with periodontitis [1]. Predisposing conditions are those that delay or prevent plaque removal and are dependent on the body’s immune response [3]. These factors include supra- and subgingival calculus, anatomical abnormalities with regard to the shape or position of the teeth, and iatrogenic factors, including restorative overhangs and subgingival margins [3]. Stress, smoking habits, diabetes mellitus, and systemic diseases are modifying factors that are responsible for the progression of the disease [3]. The nature and course of the inflammatory response is also altered. 

Next to the gut, the oral cavity is the second-largest microbiota, harboring almost 700 different bacteria [4]. Thus, the oral microbiome is the leading source of periodontitis, wherein bacterial pathogens generate an inflammatory response that marks the damage of connective tissues [5,6]. The Gram-negative bacteria responsible for causing periodontitis include *Aggregatibacter *actinomycetemcomitans**, *Porphyromonas gingivalis*, *Prevotella intermedia*, and *Tannerella forsythia* [7]. Dental plaque is an excellent example of how most naturally-occurring bacteria grow on surfaces as biofilm [7]. *Porphyromonas gingivalis* is a Gram-negative anaerobic bacteria that has been detected in 85.75% of subgingival plaques of chronic periodontitis cases [6]. 

The important role of periodontal infection as a risk factor for the development of cancer has been a cause of concern. High levels of periodontal pathogens may also predispose patients to gastric cancers as a result of being mediated by systemic infection and inflammation [8]. The Joint European Federation of Periodontology/American Academy of Periodontology Workshop in November 2012 also addressed the risk of periodontitis and its potential to cause the development of cancer. The workshop proposed that future studies should aim to fulfil the Bradford Hill or equivalent criteria [9].

## 2. Classification of Periodontitis

Periodontitis is usually seen in adults, but children and adolescents may also present with it. After much debate, the term ‘adult periodontitis’ was replaced with the term ‘chronic periodontitis’ [10]. The terms “chronic” and “aggressive” periodontitis were first introduced at the 1999 World Workshop for the Classification of Periodontal Diseases and Conditions [11]. Interestingly, the previously described types, i.e., “chronic” and “aggressive”, are now included under one single category of “periodontitis.” A consensus conference was later held in 2017 to update the earlier 1999 classification. According to earlier descriptions, in addition to earlier described types, two other different types of destructive periodontal diseases, i.e., periodontitis of systemic disease and necrotizing periodontal diseases, were also mentioned [12]. 

According to the European Federation of Periodontology report, the staging of the disease needs to consider: the (i) severity, (ii) complexity, and (iii) scoring [13]. Furthermore, there are a few essential goals described by researchers [13]. The primary goal is to classify the severity and extent of destroyed tissue because of the disease [13]. This is done with the help of radiological investigation. Regarding the second goal, clinicians should aim at assessing the complexity involved in controlling the disease and the long-term duration of this management [13]. The stages are then scored, and the severity score is primarily calculated based on interdental attachment loss. The case’s complexity is what determines the complexity score. Grading a periodontitis patient involves estimating the future risk of periodontitis progression and the possible responsiveness to standard therapeutic interventions [13]. In addition, direct and indirect evidence also need evaluation [13]. 

Rapid damage to the periodontal ligament and alveolar bone could occur in periodontitis. The extent of tissue destruction in periodontitis is generally determined by the surface of the dental plaque, host defense, and the linked risk factors. A vital characteristic of periodontitis is specific to the site of involvement, the loss of the specific periodontal pockets and their attachments, and the loss of bones that are not uniform throughout the teeth. Actual evidence does not support a different pathophysiology for chronic and aggressive periodontitis. Accordingly, the definition of a case of periodontitis largely depends on both the extent of the disease (number of affected teeth) and the severity of the disease (severity of depth of pocket, loss of clinical attachment, and loss of alveolar bone in the affected tooth) [13]. Further epidemiological studies are needed to know the etiology of the disease and its prevalence and risk factors for proper planning of adequate treatments and prevention programs.

## 3. Clinical Features of Periodontitis

The most important aspect in the management of periodontal disease is to diagnose the condition as early and accurately as possible, as extensive damage to the periodontal bone and soft tissue may be difficult to tackle. In the early stages of periodontal disease, patients hardly complain of any pain. Patients may complain of bleeding while brushing, but seldom is pain described. Clinical features of periodontitis include gum redness, changes in structure and swelling, and bleeding from the gum area. There is periodontal pocketing and attachment loss, often there is bad taste or odor, and in the most advanced forms there is tooth loss [14]. Weakening of the dental support can cause severe pain due to abscesses or tooth loss. Compared to other inflammatory conditions, periodontitis may sometimes be present without any pain [15]. Painless presentation explains the fact about why the detection may be delayed and that severe periodontitis is the main cause of tooth loss in adults [15].

The absence of pain sensation is due to the changes in periodontal nociception with regard to painless periodontal conditions [15]. Bacterial virulence, host response, altered innervation of the affected part, and suppression of the inflammation may be main factors that determine pain sensation [15].

In some cases, the disease progresses at a slower rate and there is minimal risk of periodontal function loss, while in others it may progress faster. In addition, some gingival sites within the same individual are more liable to the development of chronic periodontitis than others [16]. It was found that race/ethnicity, level of poverty, and education were associated with increased chances of developing periodontitis [16]. 

## 4. Diagnosis of Periodontitis

At present, the diagnosis of periodontal diseases is mainly based on radiographic and clinical examinations of periodontal tissues. Much is needed for the early detection, ascertainment of the severity, and prognosis of the disease, as the present tolls may be inadequate. Using clinical parameters such as bleeding on probing, probing depth and clinical attachment level, and radiographic analysis of supporting bone, proper diagnosis can be made [17,18]. Additional evidence, such as medical and family history and specific medical characteristics, also helps with diagnosis. The diagnostic process also depends on the individual skill of the examiner. Furthermore, the examination is to be repeated at regular intervals to monitor the course of the disease.

It is necessary to examine the oral cavity, measure the depth of pockets, and obtain proper radiographs to know the severity of the disease. Periodontitis biomarkers, such as lactoferrin, hemoglobin, and leukocytes in the saliva, may help in diagnosis. A research study investigated IL-1β, IL-6, MMP-8, and IL-10 levels in healthy as well as periodontitis patients and found that IL-1β and IL-6 concentrations were significantly higher in periodontitis patients [19]. This proves the importance of investigating the underlying biomarkers. 

Once diagnosed, the healthcare practitioner should immediately remove the etiological factors (microbial biofilm on the surface of teeth and gums) and advise the patient about the different risk factors involved. The reversible risk factors can be well controlled. Hence, successful management of the disease should always aim at providing prior and necessary information to the patient about its possible association with any other systemic disease [20]. 

## 5. Inflammatory Changes Occurring in Periodontitis 

In periodontitis, inflammatory changes are observed in tooth supporting tissues. Various oral pathogens, such as bacteria, fungi, and viruses, are responsible for causing inflammation. Inflammation is one of the earliest protective responses of the body against any external pathogen. While acute inflammation occurs for a few days, the chronic inflammation may last for months and years. In periodontitis, there is chronic inflammation, which is mediated by different inflammatory mediators. 

Lipopolysaccharide (LPS) is considered to be an important virulence factor of Gram-negative bacteria, and is made up of large molecules of lipids and polysaccharides. The glycolipid LPS, also known as an endotoxin, interacts with the host’s immune system. Increased levels of serum LPS result in the activation of macrophages that act as regulatory agents on the immune system of the body. Thus, macrophage-target therapy for inflammatory conditions in periodontitis may prove to be beneficial.

Periodontitis involves a severe inflammation that causes the destruction of tooth supporting apparatus. A disturbance occurs in the balance between subgingival microbiota and the host’s inflammatory response. Colonization of subgingival anaerobic bacteria is an important event that occurs in periodontitis. The microorganisms involved in periodontitis increase the expression of genes, which are involved in the immunological and inflammatory responses, cell cycle, and apoptosis. In periodontitis, genetic loci could be related to bacterial colonization. Genetic biomarkers highlight the host’s genetic variants, which are related to periodontitis. Thus, in the future, epigenetic regulation could help in the better treatment of periodontitis. 

The inflammatory mediators are responsible for the activation of various pathways that lead to bone resorption. The nuclear factor-kappa B (RANK)-RANK ligand (RANKL)-osteoprotegerin (OPG) axis is one of the molecular mechanisms regulating bone remodeling activity. By activating its receptor on the surface of pre-osteoclasts, RANK triggers their differentiation into mature osteoclasts; conversely, by binding to RANKL, OPG prevents its interaction with RANK and subsequently all the molecular events that lead to bone resorption. Under physiologic conditions, the remodeling process is characterized by the coupling of degradation of the bone matrix by the osteoclasts and its reformation by osteoblasts. When the balance switches towards alveolar bone resorption, an excess of osteoclast activity leads to a disturbance in the bone modelling, as seen in periodontitis. Lipopolysaccharides (LPS) from Gram-negative bacteria are an important agent that causes inflammatory changes and bone resorption in periodontitis. Hence, checking the osteoclastic activity certainly helps in preserving the bone mass in periodontitis.

The dysregulation between pro-inflammatory and anti-inflammatory cytokines is another molecular mechanism contributing to the periodontal tissue damage. It has been widely demonstrated that cytokines play a significant role in maintaining tissue homeostasis, and in regulating the immune response and cell signaling. During inflammatory reactions, LPS from Gram-negative bacteria elicit the production of pro-inflammatory cytokines [21]. Consistently, a higher cytokine level has been directly related to alveolar bone loss. Thus, specific cytokine targeting therapies may always prove to be beneficial.

Many inflammatory cytokines, such as interleukin (IL)-1, IL-6, IL-17, and tumor necrosis factor (TNF)-α, play an important role in the pathogenesis of periodontitis. The pro-inflammatory cytokine IL-1 promotes the activation of both Th1 and Th2 cells, which are involved in the host immune response related to the pathogenesis of periodontitis [21]. In particular, it induces cell cycling and turnover of CD4+ cells, thereby increasing the degree of expansion of naïve and memory CD4+ T cells challenged by the microbial antigen [21]. Hence, proper understanding of IL-1’s activity in CD4+ T cells is necessary for designing future drug targets [21]. IL-1 is also responsible for bone resorption and its level may directly relate to the severity of the disease. The overall amount of IL-1 in gingival fluid, which is associated with the intensity of periodontitis, decreases following periodontal treatment, thereby highlighting the importance of inflammatory markers [22]. It is also important to determine the levels of interleukin-1 receptor antagonist (IL-1ra), which may be sufficient to check the interleukin-1-induced responses in periodontium [22].

The mechanism of inflammatory changes in the bone involves various cytokines, such as (IL)-1α, IL-1β, IL-6, IL-12, and tumor necrosis factor (TNF)-α. There are certain regulatory cytokines, such as IL-4, IL-1 (RA) receptor antagonist, and IL-10. Il-6 is linked to erosion of joints, as seen in rheumatoid arthritis. Il-6 is also responsible for prostaglandin E_2_ production, and it plays a major role in the regulation of bone metabolism. IL-17 induces the secretion of IL-6, IL-8, and PGE2; hence, it is thought to play a crucial role in regulating inflammation. IL-17 is also thought to affect osteoclast activity and thereby mediate bone resorption.

The level of IL-1β may directly relate to the severity of the disease. The overall amount of IL-1β in gingival fluid, which is associated with the intensity of periodontitis, decreases following periodontal treatment, thereby highlighting the importance of inflammatory markers [22]. It is also important to determine the levels of interleukin-1 receptor antagonist (IL-1ra), which may be sufficient to check the inteleukin-1-induced responses in periodontium [22].

The pathogenesis of peridontitis is also regulated by the NLRP3 inflammasome signaling pathway [23]. A significantly higher level of nod-like receptor family pyrin-domain-containing protein (NLRP) 3 and apoptosis-associated speck-like protein containing a caspase recruitment domain (ASC) was observed in the saliva in periodontitis patients [23]. Periodontitis patients demonstrated activation of the NLRP3, which governs the conversion of pro-IL-1β into its active form [23]. Thus, the NLRP3 inflammasome signaling pathway inhibitors may also be considered as future drug targets. 

The CD4+ T cells mediate inflammation in periodontitis. CD4+ T cells are also involved in the body’s immune response against any microorganism. *P. gingivalis* could regulate the immune response via peripheral activation of T cells by enhancing the pro-inflammatory cytokine reaction [24]. Studies revealed that by causing peripheral CD4+ T helper cells to give rise to excessive amounts of pro-inflammatory cytokines, including IL-1β and IL-6, *P. gingivalis* could help in better understanding severe periodontitis [24]. Recent research related to autophagy-modulated CD4+ T cells could be helpful for designing future drug targets.

## 6. Periodontitis Associated with other Systemic Diseases

Periodontitis has been found to be associated with various systemic diseases (Figure 1). These systemic diseases are related to the gastrointestinal, cardiovascular, endocrine, respiratory, and central nervous systems of the body, and even cancers. Periodontal infections in the mouth, and their by-products, can influence the immune response outside of the mouth, thereby contributing towards the development of systemic illnesses. The periodontal region can also cause other non-oral problems, either directly or indirectly. Endotoxins have also been associated with systemic illness in many oral bacteria species, which is mainly Gram-negative anaerobic bacteria [5,6,7].

### 6.1. Cardiovascular Diseases

There is a link between periodontitis and heart disease. Individuals with the presence of periodontal disease have two to three times the risk of suffering from a heart attack, stroke, or any other serious cardiovascular event [25]. The reason attributed to such an increase is the body’s inflammation, which acts as a silent killer [25]. Interestingly, clinicians and research scientists met at a joint meeting of the European Federation of Periodontology (EFP) and the American Academy of Periodontology (AAP) and reviewed the latest scientific research related to periodontitis and atherosclerotic cardiovascular diseases (ACVD) [26]. The researchers showed that periodontitis increased the risk of a first ACVD event, (such as heart attack or stroke), and that this was independent of other known cardiovascular risk factors [26]. The risk was reported to vary according to the type of ACVD, and age and gender of the individual [26].

A meta-analysis of five comprehensive studies (86,092 patients) collectively showed that individuals with periodontal disease had a 1.14-times higher risk of coronary heart disease than those without [27]. Case–control studies with 1423 patients showed a 2.22-times higher risk of developing coronary heart disease among those with periodontal diseases compared to controls [27]. Studies have also shown there may be a link between edentulousness and serum antibodies against *P. gingivalis* and *A. actinomycetemcomitans*. DNA from periodontal pathogens such as *P. gingivalis*, *A. actinomycetemcomitans*, *P. intermedia*, and *T. forsythia* have been found in human atherosclerotic plaques, signifying that these oral pathogens pass into the body from the oral cavity and can move to remote locations [28]. More recently, studies in animal models of atherosclerosis using hyperlipidemic mice infected with *P. gingivalis* and *T. denticola* have shown that infection with these bacteria is linked with alveolar bone resorption and aortic atherosclerosis [29]. Research studies have reported that *P. gingivalis* can induce strong platelet aggregation and this may be important in the development of atherosclerotic plaques [29].

In periodontitis, *P. gingivalis* has been found in the circulating leukocytes and in atherosclerotic lesions, and this acts as a pro-atherogenic stimuli [30]. *P. gingivalis* alters the gut microbiota, thereby leading to epithelial permeability and endotoxemia, and these events cause systemic inflammation [30]. Furthermore, *P. gingivalis* from the oral site may move to the site of the blood vessels. *P. gingivalis* invades the endothelial cells of the vessels, and the resultant inflammatory changes cause endothelial dysfunction.

Patients with periodontitis may have higher chances of developing a stroke. Published reports show an association between moderate and severe stroke and periodontitis [31]. In male patients, there is a necessity to alert the physician for extensive examination, especially when other risk factors like stroke are present [31].

### 6.2. Pneumonia and Respiratory Tract Infections

Saliva and plaque in the oral cavity of individuals with periodontal disease become a possible means of storing pathogens and gradually passing them to the lower airways. Many oral pathogens have been implicated in lung infections, including *A. actinomycetemcomitans*, *Actinomyces israelii*, *Capnocytophaga spp*, *Chlamydia pneumoniae*, *Eikenella corrodens*, *F. nucleatum*, *Fusobacterium necrophorum*, *P. gingivalis*, *P. intermedia*, and *Streptococcus constellatus* [32,33,34]. Respiratory pathogens isolated from dental plaque and bronchoalveolar lavage fluid from the same patients in the intensive care unit were shown to be genetically the same, which provides evidence that dental plaque could serve as a significant reservoir for respiratory pathogens [35]. Common oral pathogens such as *F. nucleatum* and *F. necrophorum* may also result in Lemierre’s syndrome, a life-threatening condition of internal jugular vein thrombophlebitis and bacteremia that begins with pharyngitis [33]. Aspiration or oral pathogens, which result in periodontal-disease-associated enzymes in the saliva and cytokines circulating as a result of periodontitis, are the factors responsible for respiratory infections [36].

*C. pneumoniae* is well considered as a pathogen of the airway and is linked with bronchitis and chronic obstructive pulmonary disease. The pathogen has also been established in the oral cavity and could potentially be transmitted from the oral cavity to the lower airways, from where it is circulated by monocytes through the bloodstream to various organs, including the heart. Studies have shown that *C. pneumoniae* infection may lead to atherosclerotic cardiovascular diseases [37].

### 6.3. Diabetes Mellitus

In adults, diabetes mellitus (DM) manifests as hyperglycemia induced by β cells in the pancreas (type 1 diabetes) or impaired insulin sensitivity (type 2 diabetes), wherein there is a defect in insulin secretion, insulin action, or both [38]. Uncontrolled diabetes poses life-threatening consequences, which include hyperglycemia with ketoacidosis or the non-ketotic hyperosmolar syndrome [38]. DM and periodontitis have a “two-way” relationship, whereby the risk of periodontitis is much higher if glycemic control is poor [39]. A chronic infection caused by periodontitis can contribute to aggravation and an irregular inflammatory response, resulting in a blood sugar metabolic mechanism that is starved, along with higher insulin requirements. Basic bacterial and viral infections cause significant and long-term insulin resistance in patients. In diabetic mouse models with *P. gingivalis* infection, reduced gingival vascular function and increased insulin resistance were observed [40]. *P. gingivalis* had its effect on intestinal microbiota and adipose tissue inflammation, and played a role in the regulation of downstream genes (such as mTOR, S6K-1) in the branched-chain amino acids (BCAAs)-induced insulin resistance signal pathway [40]. A recent study showed that periodontitis is exacerbated by diabetes in three ways: (i) IL-17 increases periodontal dysbiosis and bacterial pathogenicity, (ii) the host’s response to the bacterial challenge is improved, and (iii) periodontal damage is increased [41]. Diabetes inhibits new bone formation by increasing the death of bone-forming cells and reducing the proliferation and differentiation of periodontal ligament stem cells (PLSCs) in osteoblasts [41].

### 6.4. Alzheimer’s Disease

The latest extensive study of oral health showed that individuals with brain injury had poor oral health parameters and a higher prevalence of chronic generalized periodontitis [42]. The brain, considered to have an immune response due to its “immunological privilege” status, may undergo various inflammatory processes that contribute to the development of Alzheimer’s disease (AD), such as complement activation, as well as the expression of cytokines and chemokines [43]. 

The neurodegeneration that occurs in AD was also shown to be a direct impact of β-amyloid plaques [44]. Chronic periodontitis has been reported to be associated with dementia [44]. In older patients with AD and periodontitis, an increase in pro-inflammatory cytokines was reported [45]. The concept that AD is a key stimulant for inflammatory neurodegeneration is supported by studies utilizing several anti-inflammatory medicines and cytokines, suggesting that non-steroidal anti-inflammatory drugs (NSAIDs) for AD may be effective in reducing the onset [46]. The mechanism regarding the relationship between periodontitis and cognitive decline is still not well understood, but there is sufficient evidence to support a role for systemic inflammation. The pro-inflammatory cytokines circulating in the blood may reach the brain via various pathways to help in the progression of neurodegenerative diseases. In periodontitis, inflammatory markers, such as C-reactive protein (CRP), and pro-inflammatory cytokines, such as Tumor Necrosis Factor α (TNFα), are increased, which eventually may be associated with cognitive decline in AD patients [47]. 

Researchers found that after a six-month follow-up period, AD participants with periodontitis had a relative rise in the pro-inflammatory state and a decrease in the anti-inflammatory state [47]. The pro-inflammatory state and the prevalence of IgG antibodies against *P. gingivalis* were related [47]. *P*. *gingivalis* is an important pathogen involved in the etiology of periodontitis. Experimental studies on mice showed that *P. gingivalis* and gingipains in the brain play a major role in the pathogenesis of AD [48]. Oral *P. gingivalis* infection in experimental mice results in brain colonization and increased production of Aβ_1–42_, which is a component of amyloid plaques [48]. *P. gingivalis* may enter the brain through various pathways and result in damage to the brain, thereby affecting cognitive functions. 

Recently, researchers have also focused on the role of *T. denticola* in AD. *T. denticola* is another pathogen associated with periodontitis. According to published studies, *T. denticola* could invade oral squamous cell carcinoma cells (OSCC) and actively encourage both in vitro cell growth and in vivo tumor formation [49]. This process was associated with the upregulation of TGF-β1, -β2, and -β3 mRNA and the activation of the TGF-β pathway [50]. *T. denticola* can alter the host immune response in AD. Hyperphosphorylation of tau protein also causes resultant neurodegenerative changes in the brain. 

### 6.5. Colorectal Cancer

The host immune response is altered by many microorganisms. *F. nucleatum* is associated with periodontal disease. Colorectal carcinoma (CRC) is the fourth most important cause of cancer mortality globally and has been linked to an increased incidence of *F. nucleatum* and *Clostridium difficile* in the intestinal microbiota of patients [50]. There are reports of *Fusobacterium* spp., *Campylobacter*, and *Leptotrichia* related to CRC [51]. Interestingly, a research study reported a strong association between oral anerobic bacteria and colorectal cancer [52]. It has been reported that even in the absence of any etiological role, any microbe or microbial signature with tumor specificity may be important for diagnosis and risk assessment [52]. It is thought that the oral *F. nucleatum* may migrate to the human intestinal tract and colonize it, leading to malignant inflammatory infections. *F. nucleatum* was reported to promote tumor development by inducing inflammation and the host immune response in hosts with CRC [53]. As a result of inflammation, *F. nucleatum* creates a microenvironment that favors tumor growth [53]. Furthermore, *F. nucleatum* has been reported to suppress immune cells such as macrophages, T cells, and NK cells, thereby causing immune suppression of the gut [53].

### 6.6. Adverse Pregnancy

Due to hormonal modifications in pregnant women, they are more predisposed to gingivitis and periodontitis [54]. The level of cytokines also increases during pregnancy. During pregnancy, the periodontal pathogens can spread to the amniotic cavity and affect the placental tissue, thereby causing maternal–fetal complications [54]. High levels of periodontal pathogens and decreased maternal IgG antibody response to periodontal microorganisms were reported in pregnant women [55]. This could predispose patients to pre-term labor and low birth weight [55].

Bacterial pathogens, antigens, endotoxins, and pro-inflammatory cytokines formed during periodontitis disease can cross the placental barrier, disrupting the maternal unit, which may contribute to the negative consequences of pregnancy [56]. *F. nucleatum* is one of the commonest communal oral pathogens established in numerous placental and fetal soft tissues [57]. *F. nucleatum* was reported to cause chorioamnionitis, preterm birth, stillbirth, neonatal sepsis, and pre-eclampsia [57]. The microorganism, *F. nucleatum,* was found in the amniotic cavity of women with preterm labor [58]. The organism may travel from the mother’s oral cavity to the uterus via a hematogenous route. Interestingly, *F. nucleatam* is frequently isolated from amniotic fluid and bone marrow in cases of premature labor and newborn sepsis, which is consistent with its more aggressive nature [59,60]. Furthermore, *F. nucleatam* is often found in intrauterine infections with other oral subspecies, possibly of the same infectious origin, suggesting co-migration from the oral cavity [61,62]. Other oral pathogens, such as *P. gingivalis* (and its endotoxins), have also been found in the placentas of patients with premature delivery [63]. *P. gingivalis* causes the disruption of cytokine expression, increases cell proliferation, and suppresses apoptosis [64]. These events could be responsible for the disruption of homeostasis related to the placenta.

A study found that Gram-negative bacterial oral endotoxins like LPS could cause a deleterious effect on the fetus [65]. Hence, it is important to check these endotoxins in periodontal diseases in pregnant women. A research study was conducted on microbiological and serological markers of periodontitis associated with conception [66]. The same study also found that *P. gingivalis* may interfere with conception [66]. Hence, it is important that women in a pregnancy state should have regular oral examinations in order to rule out periodontal disease. Many studies have reported that periodontal disease is associated with an increased risk of adverse pregnancy outcomes, such as preterm birth and low birth weight [67,68]. 

### 6.7. Multiple Myeloma

Multiple myeloma (MM) is a clonal plasma cell proliferative disorder that is characterized by the abnormal increase of monoclonal paraprotein, thereby leading to specific end-organ damage [69,70]. In this disease, there is monoclonal gammopathy of undetermined significance and monoclonal immunoglobulin is detected in the blood or urine [69]. Although the exact etiology of MM is unknown, it is associated with genetic abnormalities in oncogenes [69], alcohol consumption, obesity, and environmental causes such as insecticides, organic solvents, and radiation exposure [71]. The disease is mainly characterized by anemia, a higher risk of infection, hypercalcemia, renal failure, and bone lesions, and is the most common hematological disorder that causes severe bone pain and fractures. The most affected areas in the body are the pelvis, spine, ribs, and skull. MM is diagnosed by a complete analysis of the proteins in the blood and urine, which can accurately reveal the monoclonal component. In addition, the detection of bone lesions should be determined by a radiological examination of the total skeleton. The principal clinical indicator of the disease is initiated in the oral cavity in some MM patients [72]. Oral cavity lesions can be the first sign of recurrence or progression of MM [72]. The maxillary pain is highly suggestive of MM and clinicians need to be aware of that [73]. MM patients may have oral manifestations of periodontitis and it is necessary to rule out any differential diagnosis [74]. Soft-tissue amyloid deposits [75] and external dental root resorption [76] are seen in MM patients. Even though excisional or incisional biopsy remains the gold standard for diagnosis of myelomatous lesions, another suitable option is fine needle aspiration (FNA) biopsy, especially in the early diagnosis of a suspicious mass or lesion [74]. There are a few advantages to FNA biopsy, and they include ease of execution, lower complication rate, and rapid diagnosis [77,78]. General and oral manifestations as well as signs of MM are shown in the schematic diagrams in Figure 2 and Figure 3. 

### 6.8. Cancer in Different Parts of the Body

The oral microbiome of the periodontal region plays a significant role in the inflammatory response. In patients with periodontal infections, there is a high level of colonization of periodontal pathogens, and this predisposes patients to the development of gastric cancer [79]. Periodontitis is primarily caused by the oral microbiota. Additionally, bacterial infections may cause periodontitis, which is characterized by an inflammatory response that leads to the destruction of bone tissue. Oral hygiene and proper control of chronic inflammatory conditions may prevent cancer [80]. *A. actinomycetemcomitans*, *P. gingivalis*, and *T. forsythia* are among the etiological contributors of periodontitis in the oral microbiome. Microbial communities associated with cancer may be important for diagnosis and prognosis and future studies are needed for a better understanding of the dysbiotic tissue microbiome [81]. A previous research study found oral pathogens such as *Fusobacterium nucleatum*, *Porphyromonas gingivalis*, *Treponema denticola*, and *Tannerella forsythia* to be associated with gastric cancer [82]. *F. nucleatum* followed by *T. forsythia* were the most abundant bacteria associated with gastric cancers [82]. Certain types of human oral bacteria, such as *F. nucleatum* or bacteroides, have also been reported to be associated with periodontitis, appendicitis, and colorectal cancer [83]. The microorganisms found in periodontitis may cause inflammation and promote carcinogenic processes.

There is a positive link between periodontal disease and non-Hodgkin’s lymphoma (NHL). Researchers reported a 31% higher risk of NHL in periodontitis patients [84]. Interestingly, the same study found that tooth loss was inversely related to NHL among those individuals without periodontal disease [84]. A previously published meta-analysis found tooth decay to be a risk factor for esophageal cancer [83]. Inflammation was the main cause in periodontitis, and this was linked to cancer. Furthermore, oral microorganisms were reported to produce greater amounts of nitrosamine, which was associated with the development of cancer of the esophagus, oral cavity, and pharynx [85,86]. Bone loss in periodontitis is also a risk factor for the development of oral cancer [83]. 

Significantly, bacterial organisms, such as oral *Streptococcus intermedius* (*S. constellates*, *S. oralis*, *S. mitis*, *S. sanguinis*, *S. salivarius*), have remained out-of-the-way from cervical lymph nodes in patients with oral cancer [87,88]. Shortly after infection, *Porphyromonas gingivalis* infection stimulates PI3K/Akt signalling, which blocks apoptosis and encourages epithelial cell survival and growth. *P. gingivalis* aids cancer cells in sustaining their growth and lifespan by blocking p53. Gingipains, the main virulent components of *P. gingivalis*, bind to and convert pro-MMP9 into MMP9 to facilitate invasion. Additionally, *P. gingivalis* modifies the immunological response via cytokines and elevates the production of B7-H1 and B7-DC receptors, which lead to activation-induced death in activated T lymphocytes [89]. 

Gingival crevicular fluid (GCF) is a target for identifying effective diagnostic markers of periodontitis because it is located in the periodontal pocket where inflammation occurs and reflects the changes in the periodontal microenvironment. TGF-β1, TGF-β2, and TGF-β3 are expressed in different epithelial, hematopoietic, and connective tissue cells [90]. TGF-β is also present in gingival crevicular fluid [90]. Researchers have reported the importance of TGF-β1 gene expressions and TGF-β1 protein levels in gingival crevicular fluid [90]. TGF-β1 was reported to exhibit pro-inflammatory and anti-inflammatory properties [90]. TGF-signaling is important for the progression of cancer. Furthermore, through the TGF-signaling pathway, many oral pathogens support cancer progression. Many drug resistance cases in cancer could also be related to TGF-signaling.

The formation of chemical compounds by bacteria during certain biological reactions may also be important for the development of cancer. Oral microbial cells catalyze the formation of N-nitroso compounds from the precursor’s nitrite and amines, amides, or other nitrosatable compounds [91]. It was also reported that various species of bacteria encompass strains capable of catalyzing nitrosation, and an ideal example is *Escherichia coli* [91].

Periodontitis is associated with oral neoplasms [92]. It was found that both periodontal disease and smoking were significant factors in the etiology of tumors and pre-cancerous lesions in the oral cavity [92]. In addition, in periodontal ailments, carcinogens from smoking and alcohol usage pass into the primary tissues [92]. It has been proposed that the microbiome also stimulates cancer progression via different signaling pathways [93]. Microorganisms are capable of producing the toxin, colibactin, which may mediate this signaling pathway [93]. In addition, colibactin is responsible for tumor growth and progression. Hence, it is important to screen all periodontitis patients for any underlying oral lesions and cancers. 

## 7. Periodontal Microbial Carcinogenesis: Underlying Mechanisms

Chronic inflammation is considered to be a risk factor for malignant mutations in various tissues, such as the oral cavity, head and neck region, esophagus, stomach, liver, colon, cervix, uterus, and urinary tract and lungs [94]. The dysregulation of immunity and chronic inflammation caused by various oral microbes have a role in the cancer cells. 

It was suggested that the stimulation of the inflammatory process and the presence of cell-stimulating signals are responsible for creating a proper environment for cell proliferation and differentiation [95]. Chronic inflammation can encourage cell proliferation and mitogenic activity through the activation of signaling pathways, such as Ras/Raf/MEK/extracellular signal-regulated kinase (ERK) [96]. An injury leads to tissue and cell damage, and there is activation of anti-apoptosis signaling pathways in the affected cells [97]. Moreover, the immune cells of the lymphoid and myeloid lineages are attracted to the site of injury [97].

It has been suggested that persistent infections are capable of damaging DNA in cells that are spread by the creation of toxic materials, such as volatile reactive oxygen species (ROS) and reactive nitrogen intermediates (RNI), by inflammatory cells [98]. As a result, DNA destruction happens as a consequence of tissue redevelopment, and proliferative cells go through permanent genomic alterations [99]. 

Cytokines and chemokines play critical roles in tumor initiation and progression [100]. Several inflammatory mediators may take part in the initiation and progression of cancer [101]. Marrow organisms can also lead to the development of cancer. Hepatitis B virus (HBV) and hepatitis C virus (HCV) in hepatocellular carcinoma can be linked to oral lesions. Some of the basic fundamental reasons that underlie HCV-linked liver cancer include improved hepatocyte proliferation, involvement of the immune and inflammatory responses, and genomic mutations [101]. The inflammation-associated epithelial–mesenchymal transition is important for the development of cancer [101]. Inflammation has been reported to be associated with metastasis [101]. High expression of E6 and E7, which play a significant role in microbial-infected cancers, has been observed during HPV infection [102]. In addition, HPV disturbs the early stages of the immune reaction, which comprises the expression of TLRs and cytokines—which significantly affects the detection of HPV [103]. Furthermore, the occurrence of HPV E6/E7 mRNA in the periodontium may provide support for the hypothesis that periodontal soft tissue acts as a reservoir of dormant HPV infection [104]. The deep periodontal pocket is an established site for viral infections, such as HPV [105], Epstein-Barr virus (EBV) [106], and Herpes simplex virus (HSV) [107].

## 8. Role of microRNA in Periodontitis

MicroRNAs (miRNA) are a group of small non-coding RNAs that negatively regulate protein expression and are involved in different physiological and pathological mechanisms. miRNAs are responsible for the initiation and progression of cancer. The miRNAs act through various pathways. Overexpression of miRNAs in progressive leukoplakias was evidence of involvement of miRNAs with cancers [108]. The researchers showed that the identified miR signature could distinguish between progressive and non-progressive leukoplakia and thus could be used as important biomarkers [108]. 

The altered expression of miRNA contributes to many cellular immune responses. miRNA expression abnormalities in biofluids such as serum, saliva, and gingival fluids of patients with periodontal pathophysiology have been reported [108,109]. For instance, miRNA-1226-5p has been proposed to be a miRNA biomarker for periodontitis [110]. miRNAs are stable in an RNase-rich environment and can be detected in fluids. The identification of new epigenetic biomarkers like miRNAs certainly help in improving the clinical diagnosis and prognosis of periodontitis [110]. 

miRNAs regulate genes that are responsible for the development of bone. Through the Wnt, bone morphogenetic protein, and Notch signaling pathways, miRNAs influence the development of alveolar bone lineage and the production of new bone in periodontal fibroblasts [111]. When periodontal tissue integrity is lost due to periodontal disease, miRNAs play a critical role in the formation, homeostasis, and maintenance of periodontal tissue [111]. miRNA therapies can be used to check the upregulated miRNAs that are linked to osteoclast development. miRNAs also contribute to bone morphogenesis and osteoclastogenesis (OsteomiRs) [112].

The expression of miRNA-21 is upregulated in *P. gingivalis* LPS-stimulated macrophages [113]. The miR-21 mimic inhibits pro-inflammatory cytokine production by macrophages, while miR-21 deficiency increases pro-inflammatory cytokine production [113]. It was found that the absence of miR-21 elevated PDCD4 expression, NF-κB activation, and pro-inflammatory cytokine production [113]. The NF-κB pathway plays a crucial role in cytokine synthesis and pro-inflammatory signaling. miRNA-21 has a protective role on the progression of periodontitis [113].

It was shown that miRNA-146 suppresses pro-inflammatory cytokine secretion by regulating the production of INA-1, IL-1, IL-6, and TNF-associated with the IL-1 receptor [114]. The miRNA let-7i was expressed in periodontitis and found to influence the Toll-like receptors 4 and promote bacterial aggression [115]. It may be mentioned that oral microbial pathogens through Toll-like receptors activate the NF-κB pathway and this contributes to inflammation, which is a key component in periodontitis infection. 

By controlling both neutrophil adherence and the stability of chemokine mRNAs, miRNAs influence neutrophil function and migration from blood capillaries into inflamed tissues [116]. The dysregulation of miRNAs can be induced by various bacterial components. Cellular miRNA levels change in many diseases, including periodontitis, and the miRNA profile may be important for diagnosis and treatment [117,118]. 

## 9. Conclusions

In this present review, we discussed periodontitis and its association with systemic diseases. Periodontitis is a chronic inflammatory disease. Various microorganisms are present in the oral cavity of periodontitis patients. In periodontitis, inflammation is the key component for the spread of microorganisms from the oral cavity to other organs. We discussed the role of cytokines as mediators of inflammation and their involvement in different pathways. There is an interaction between the microorganism present and the host’s immune response. According to researchers, regarding periodontitis, there is a feedforward loop between host and microbiota variables that is responsible for the onset and persistence [119]. Knowledge regarding oral polymicrobial synergy and dysbiosis is also important for a proper understanding of the pathogenic mechanisms [119]. To summarize, differential diagnosis and the association of periodontitis with other systemic diseases require a cautious approach. Hence, it is important to screen periodontitis patients with associated systemic diseases and understand the underlying mechanism of periodontal microbial carcinogenesis, which can be fatal. There is a need to investigate all possible genetic-, protein-, microorganism-, and host response-related biomarkers for the detection of periodontitis at an early stage. The review also discussed the latest concepts on the staging and grading of the disease and the important role of miRNA as a marker for prognosis, and its diagnosis and treatment. This review may be beneficial to health professionals, clinicians, and dental health professionals who are involved in treating periodontitis.

## Figures and Tables

**Figure 1 biomedicines-10-02659-f001:**
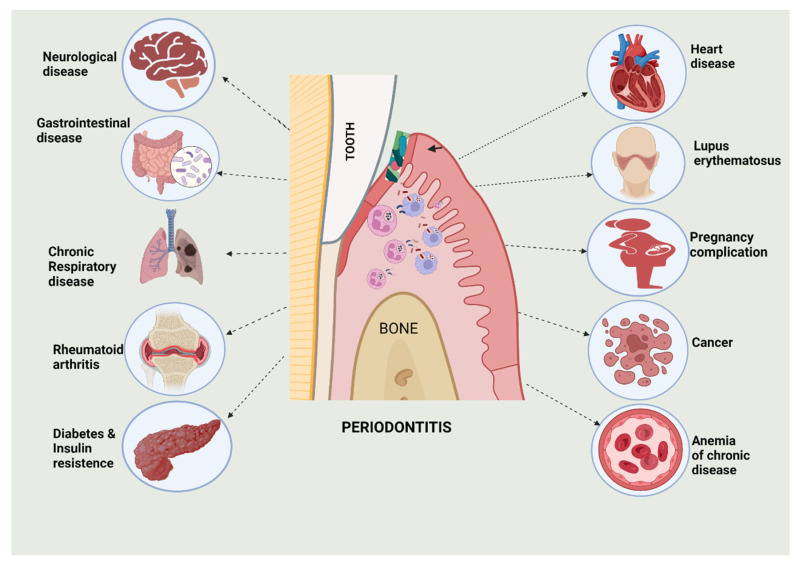
Representation of diverse systemic diseases and their relationship with periodontitis. This figure was created with Biorender.com (accessed on 9 September 2022).

**Figure 2 biomedicines-10-02659-f002:**
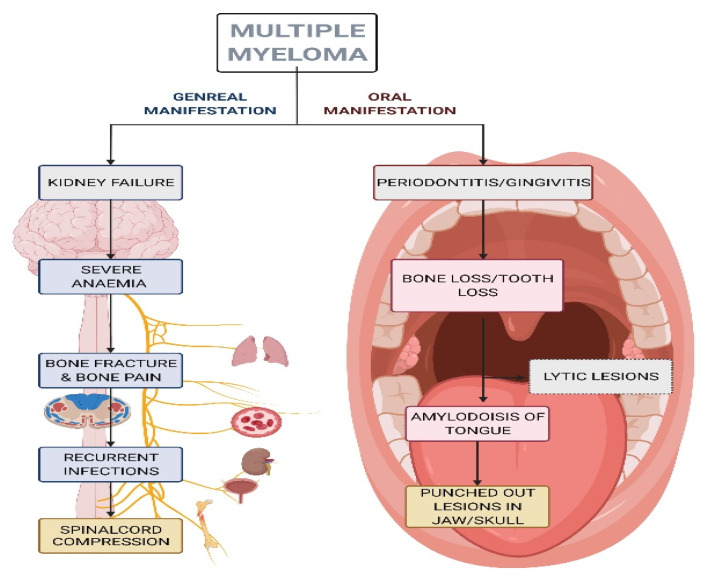
Schematic diagram showing general and oral manifestations of multiple myeloma. This figure was created with Biorender.com (accessed on 9 September 2022).

**Figure 3 biomedicines-10-02659-f003:**
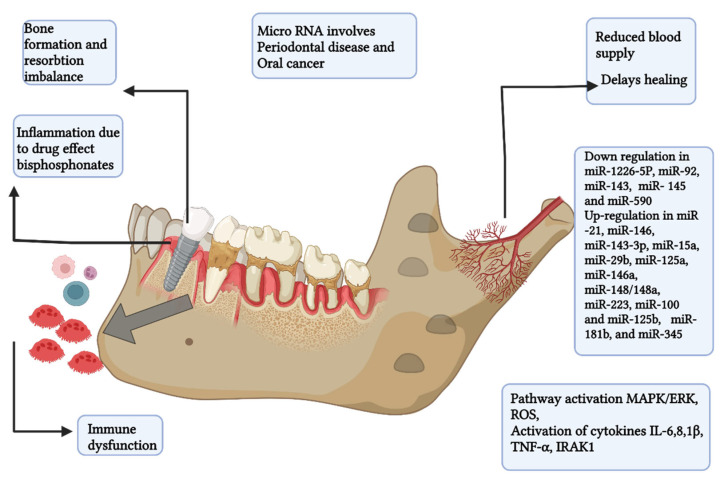
Schematic diagram showing events in the oral cavity. This figure was created with Biorender.com (accessed on 9 September 2022).

## Data Availability

Not applicable.

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
