# Peer review of "Periodontitis and Its Inflammatory Changes Linked to Various Systemic Diseases: A Review of Its Underlying Mechanisms"

_biomedicines, 2022, doi:10.3390/biomedicines10102659_

Round 1
Reviewer 1 Report
Dear Authors,
your paper looks like a story not like
a scientific article: there is no aim, material,
method, discussions, etc.
It cannot be seen any “ molecular perspective”
as the title would suggest. It is completely inappropriate
for publication in this journal.
Author Response
It is a narrative review. Hence, no specific subheadings on methods, discussion were added.
We are sorry if we could not impress the reviewer.
We tried our best to change the title to "Periodontitis and its Inflammatory Changes Linked to Various Systemic Diseases: Underlying Mechanisms.
We added more text on the molecular aspects. Changes are shown in RED color.
Reviewer 2 Report
General comments: This review manuscript discuss the epidemiology, clinical features, diagnosis, complications, and treatment of periodontitis. In addition, the association of chronic inflammation found in periodontitis with various other systemic diseases including cancer and microRNAs role was also discussed in length. Overall this review comprised periodontal disease and its link with several systemic diseases including oral cancer. Authors included three schematic diagram (figures) in this review. First the review needs significant revision with regard to correct grammar, incorrect statements, sentence structure, bacterial nomenclature, spoken language, etc. There were simply too many examples of this to list here. There are several sentences require clarity and some of them are in spoken language. This review will be beneficial if it is edited by English language editorial service. This reviewer incorporated all the comments in the pop-up or sticky note in the pdf file.
Specific comments: Authors should check for recent articles for newer findings in the systemic diseases linked with periodontal disease. Authors also requested to check the references appropriate for the statement as few are incorrect references for the statement mentioned. Newer bacterial name and all bacterial names should be in italics. Authors should change gram-negative to Gram-negative as Gram is inventor name. Authors check Figure 3 labeling errors.

Author Response
English language was rectified.
We thank the reviewer for the corrections. We incorporated all such corrections.
References were rectified.
Fig.3 was checked again.
'Gram' spelling was corrected in upper case.
Reviewer 3 Report
Bhuyan et al. provide for review manuscript entitled: “Periodontitis and its Inflammatory Changes Linked to Various Systemic Diseases: An Update.” This review focuses on describing the clinical presentation, etiology, treatment, pathogenesis, and connection between periodontal disease and systemic diseases. The authors should provide the following changes before this paper will be suitable for publication:
Major revisions:
1) The authors should dedicate more time to the “Inflammatory Changes in Periodontitis” lines 119-137. This section is extremely superficial. It would be beneficial to explore specific cytokines present in chronic periodontitis (mucosal vs. immune cell derived), as well as how individual periodontal pathogens influence these pro-inflammatory cytokines (direct proteolytic cleavage like with P. gingivalis gingipains vs. differential regulation of gene expression).
2) It’s unfortunate that there is a lack of basic science representation in this paper. The authors generally present associations without going into any specific mechanisms that have been established. For example: Alzheimer ds and P. gingivalis (Dominy SS et al Porphyromonas gingivalis in Alzheimer's disease brains: Evidence for disease causation and treatment with small-molecule inhibitors. Sci Adv. 2019), T. denticola and cancer (Peng RT, Sun Y, Zhou XD, et al. Treponema denticola Promotes OSCC Development via the TGF-β Signaling Pathway. Journal of Dental Research. 2022), and a plethora of papers on P. gingivalis influencing proliferation, apoptosis, and epithelial to mesenchymal transition.
Minor Revisions:
1) Line 52- Change actinobacillus to aggregatibacter
2) Line 79- Unnecessary hyphen
Author Response
We thank the reviewer for the comments.
“Inflammatory Changes in Periodontitis” section has been re-written. Cytokines were mentioned.
Specific mechanism related to AD was dealt in detail. We added more facts with 2 references suggested by the reviewer (Ref. 48 and 49).
P. gingivalis was discussed in detail. We also discussed T. denticola in detail.
All other errors were rectified.
We tried our best to answer all queries.
Round 2
Reviewer 1 Report
Dear Authors,
there is a correlation that have been missed in the paper, between periodontal disease and ischemic stroke patients. You should read the paper: Ioana Stănescu, Adriana Elena Bulboacă , Iulia Cristina Micu, Sorana D. Bolboacă , Dana Gabriela Festilă, Angelo C. Bulboacă, Gyorgy Bodizs, Gabriela Dogaru, Paul Mihai Boarescu, Aurel Popa-Wagner, Alexandra Roman. "Gender Differences in the Levels of Periodontal Destruction, Behavioral Risk Factors and Systemic Oxidative Stress in Ischemic Stroke Patients: A Cohort Pilot Study." MDPI, J. CLIN. MED. 2020, 9, 1744; doi:10.3390/jcm9061744. and include it in the reference list. You will find a correlation to the title.
Also, to increase the clinical application of the paper in everyday practice, at the end of Diagnosis section, it would be included the necessity of informing the patient about the association between existing periodontitis and other systemic diseases that patient could suffer from.. For this, you should check the following article: Preoteasa CT, Dumitrache A, Enache M, Iosif L, Preoteasa E. "Patient's information on medical aspects". Romanian Journal of Oral Rehabilitation, vol.10. no.1, January-March 2018.
Author Response
There is a correlation that have been missed in the paper, between periodontal disease and ischemic stroke patients. You should read the paper: Ioana Stănescu, Adriana Elena Bulboacă , Iulia Cristina Micu, Sorana D. Bolboacă , Dana Gabriela Festilă, Angelo C. Bulboacă, Gyorgy Bodizs, Gabriela Dogaru, Paul Mihai Boarescu, Aurel Popa-Wagner, Alexandra Roman. "Gender Differences in the Levels of Periodontal Destruction, Behavioral Risk Factors and Systemic Oxidative Stress in Ischemic Stroke Patients: A Cohort Pilot Study." MDPI, J. CLIN. MED. 2020, 9, 1744; doi:10.3390/jcm9061744. and include it in the reference list. You will find a correlation to the title.
We thank the honorable reviewer for the comment. We have added the facts and cited this reference (Reference 31, page 5).
The revised paragraph may be read as- “Patients with periodontitis may have more chances of developing stroke. Published reports showed an association between moderate and severe stroke and periodontitis [31]. In male patients, there is a necessity to alert the physician for extensive examination especially when other risk factors like stroke are present [31].”
Also, to increase the clinical application of the paper in everyday practice, at the end of Diagnosis section, it would be included the necessity of informing the patient about the association between existing periodontitis and other systemic diseases that patient could suffer from.. For this, you should check the following article: Preoteasa CT, Dumitrache A, Enache M, Iosif L, Preoteasa E. "Patient's information on medical aspects". Romanian Journal of Oral Rehabilitation, vol.10. no.1, January-March 2018.
We went through the paper which was stated by the reviewer. The published paper did not specially point to periodontitis and its association with any other systemic disease but still then, we comply with the suggestion, and we include this suggested reference (Reference 20, page 3)
The sentence was revised as follows- “Hence, successful management of the disease should always aim at providing prior and necessary information to the patient about its possible association with any other systemic disease [20].”
Reviewer 2 Report
Authors have improved the review article. However, authors have to delete spoken words such as checked, death, produce, etc. and replace with scientific words. One reference repeated again. Authors add italics to bacterial names.

Author Response
Authors have improved the review article. However, authors have to delete spoken words such as checked, death, produce, etc. and replace with scientific words. One reference repeated again. Authors add italics to bacterial names.
We have rectified the spoken words.
In scientific statements, ‘death’ word was used. At other places, it was replaced with the word ‘mortality.’
The word ‘produced ‘was changed to ‘formed.’
‘produce’ was changed to ‘gives rise to.’
Reference 10 was duplicated as reference 14. Sorry for the error. We changed it to reference 14.
Gasner NS, Schure RS. Periodontal Disease. [Updated 2022 May 8]. In: StatPearls [Internet]. Treasure Island (FL): StatPearls Publishing; 2022 Jan-. Available from: https://www.ncbi.nlm.nih.gov/books/NBK554590/
We have changed all names of bacteria to italics. Even an abbreviation list was added at the end.
Reviewer 3 Report
Bhuyan et al. provide for review manuscript entitled: “Periodontitis and its Inflammatory Changes Linked to Various Systemic Diseases: Underlying Mechanisms" for publication. The authors focus on the relationship between periodontal disease and systemic conditions, such as oral cancer, Alzheimers, and Diabetes. The authors have improved the publication since the previous submission, but I would still recommend the following change:
1) The section on pathogenesis of periodontal disease is weak. There are many papers establishing P. gingivalis, T. denticola, and other organisms in periodontal disease, and they are hardly mentioned. The microbiome is a significant player in perio and should be expanded on. Expanding on these organisms will also make more sense as you connect them to the pathogenesis in other systemic conditions. You can use the following paper for reference:
Lamont RJ, Koo H, Hajishengallis G. The oral microbiota: dynamic communities and host interactions. Nat Rev Microbiol. 2018 Dec;16(12):745-759. doi: 10.1038/s41579-018-0089-x. PMID: 30301974; PMCID: PMC6278837.
Author Response
Bhuyan et al. provide for review manuscript entitled: “Periodontitis and its Inflammatory Changes Linked to Various Systemic Diseases: Underlying Mechanisms" for publication. The authors focus on the relationship between periodontal disease and systemic conditions, such as oral cancer, Alzheimers, and Diabetes. The authors have improved the publication since the previous submission, but I would still recommend the following change:
1) The section on pathogenesis of periodontal disease is weak. There are many papers establishing P. gingivalis, T. denticola, and other organisms in periodontal disease, and they are hardly mentioned. The microbiome is a significant player in perio and should be expanded on. Expanding on these organisms will also make more sense as you connect them to the pathogenesis in other systemic conditions. You can use the following paper for reference:
Lamont RJ, Koo H, Hajishengallis G. The oral microbiota: dynamic communities and host interactions. Nat Rev Microbiol. 2018 Dec;16(12):745-759. doi: 10.1038/s41579-018-0089-x. PMID: 30301974; PMCID: PMC6278837
We thank the honourable reviewer for the valuable comments. P. gingivalis and P. denticola are included under each section under different diseases rather being described in any particular single section. Extra sentences with facts were also incorporated. All these portions in the text were highlighted in RED colour for easy interpretation by the honorable reviewer.
The suggested reference (Lamont et al.) was from a review article; we have added this reference (Reference 117, page 12).
We complied with all suggestions.